# Topic and Description Reasoning Generation based on User-Contributed Comments

## Abstract

We propose Topic and Description Reasoning Generation (TDRG), a text inference and generation method based on user-contributed comments with large language models (LLMs). Unlike summarization methods, TDRG can infer the topic according to comments contributed by different users, and generate a readable description that addresses the issue of the lack of interpretability in traditional topic modeling for text mining. In this paper, we adopted zero-shot and fine-tuning methods to generate topics and descriptions for comments. We use a human-annotated YouTube comment dataset to evaluate performance. Our results demonstrate that the potential of large language models of reasoning the topic and description. Generated topic titles and descriptions are similar to human references in textual semantics, but the words used are different from those of humans.

## 1 Introduction

With the rapid development of Internet, user-generated content has grown exponentially, flooding social media platforms, forums, and websites with an unprecedented volume of data. The vast majority of user-generated content is generated on social media platforms. People who frequently use social media platforms share their lives and interact with others via various formats, including text, image, video, audio, etc. However, these data formats are usually unstructured. For researchers, it is a challenge to extract meaningful insight from a massive volume of unstructured data. Thanks to recent advances in natural language processing, many studies started to explore solutions to handle unstructured data with large language models. GPT-4o (https://openai.com/index/hello-gpt-4o/), which is a powerful large language model released in May 2024 by OpenAI, can reason across audio, vision, and text in real time. The outstanding and cross-modal abilities of GPT-4o encourage us to deal with complex user-generated content on social media.

YouTube, which is the largest video-sharing and social media platform, is primarily focused on video and audio content. People can discuss videos and interact with YouTubers, the video creators on YouTube, by commenting under the videos. In previous studies, researchers employed the natural language processing techniques to process massive volume of text comments, including topic modeling and text summarization (Wu & Li, 2019; Chakraborty et al., 2019). The purpose of these methods is to help humans to quickly gain insights from massive textural data. It is important for YouTubers to understand what their audiences' thoughts are. Through audience feedback, YouTubers can discover their preferences and produce more engaging content that caters to them. However, users' comments encompass a multitude of aspects, including the content of the videos themselves, the personalities of the YouTubers, and other unrelated matters such as "spam comment". Our study focused on these comments related to the video content instead of the others. We proposed a Topic and Description Reasoning Generation (TDRG) approach to automatically generate the topic title and reasoning the description about the topic.

The TDRG method comprises three main components: principles, comments, and a large language model. The core component is the large language model, which plays a crucial role in generating high-quality topics and descriptions. In this paper, we observe that as the model size increases, its performance improves significantly, especially in zero-shot learning tasks. This is because smaller models do not adhere well to the principles of constraint. As a result, the generated topics differ significantly from human-written ones, and the descriptions are even less aligned. To improve the

results, we used the LoRA method to fine-tune the Llama-3.1-8B-Instruct quantized model with human-annotated comments collected from publicly available YouTube videos. We demonstrated that the topics and descriptions generated after fine-tuning are significantly better than those produced before.

The main contributions of this paper:

1. We formalize the task of topic and description reasoning generation and build an automation approach to generate the topic and description based on user-contributed comments.

2. We found that providing additional context, such as captions, can sometimes interfere with the output of large language models (LLMs).

3. We demonstrate that fine-tuning model using human-annotated comments significantly improves the quality of generated topics and descriptions.

## 2 RELATED WORK

### 2.1 TOPIC MODELING

There exists a number of topic modeling approaches in text mining, which are used to identify underlying themes, patterns, and structures in large volumes of text data. Blei et al. (2003) proposed a "Latent Dirichlet Allocation" (LDA), which is probabilistic model for collections of discrete data. It is an unsupervised approach to discover latent topics in text data. LDA has also been extended and modified to improve its performance due to its defects, including the inability to perform well in short text and ignoring semantic similarity. MSTM and EXTM (Ma et al., 2012) consider the relationships between comments and news articles and assume that each comment has one topic due to the short length of most comments. Lu et al. (2020) proposed a subject word feature extraction method LDA-SLP (LDA-word vector Similarity & Location & Part of speech). The weight of the candidate subject words is determined by the similarity between candidate subject words and article category label, part of speech, and word position of the candidate subject words. However, these approaches focus on optimizing topic detection and topic clustering instead of topic explanation. This activates us to study and understand where the topic comes from and how to explain it, which we call topic reasoning.

### 2.2 TEXT GENERATION

Since the launch of ChatGPT in 2022, Natural Language Generation (NLG) has become popular research. ChatGPT is a large language model (LLM) was built by OpenAI, which demonstrated a powerful generative ability in various natural language processing tasks. With LLMs' ability, many studies have explored complex textual structures such as table, SQL (Structured Query Language) (Gong et al., 2020; Dong et al., 2023) .In recent studies, a lot of researchers attempt to unleash the LLMs' generative abilities though prompt engineering, which is a zero-shot learning with large language methods, and fine-tuning (Sahoo et al., 2024; Hu et al., 2021). Prompt engineering is a straightforward method involved crafting various input texts, referred to as prompts, which influenced the model's output. Otherwise, fine-tuning represents a sophisticated method involved in preparing domain-oriented datasets for training and teaching LLMs' how to generate the superior output via human instruction and human feedback. Fine-tuning is classified as a transfer learning to enhance the system's capability in domains. However, our TDRG method aims to tackle the novel topic reasoning task. In this task, we provide comments from YouTube videos to a large language model, which is then required to infer the topic from these comments, generate an appropriate topic title, and even produce an explanatory description of the topic.

## 3 METHOD

In this section, we formalize the task of Topic and Description Reasoning Generation (TDRG) addressed in this paper, and then introduce our data, human-annotate process, and TDRG components.

## 3.1 TASK FORMALIZATION

At first, we divided our task of generating topics and descriptions into two subtasks: topic generation and topic description generation. These two subtasks are interdependent, as the output of one influences the other. In essence, the topic is derived from the general subject of the comments, while the description elaborates on the discussion within those comments under the topic. Typically, the topic is a concise phrase or sentence, around 15 words in length, with the description providing further explanation, around 150 words in length.

## 3.2 DATA COLLECTION

There is no public dataset suitable for our tasks. Therefore, we crawled the comments from YouTube videos and wrote topic and description manually. We collected user-contributed comments from YouTube public videos via YouTube Data API.

In this paper, we mainly focus on Chinese comments instead of other languages.

Each video randomly selected 100 comments in order to ensure a consistent sample size.

We crawled 125 videos from 25 representative YouTubers in Taiwan. Their channels mainly speak Chinese and audiences also comment with Chinese. Therefore, we calculated the number of Chinese words in each comment and remained top 25 videos, where text exceeded five Chinese word. There were finally 1,972 comments for the next human-labeling phrase.

## 3.3 HUMAN ANNOTATIONS

According to the task formalization for topic and description reasoning generation, We formulated principles for writing topics and descriptions respectively. We manually clustered the comments and wrote the topics and the descriptions for each clustered set based on our writing principles. Finally, the human-annotated YouTube comment dataset contains 112 clusters derived from 25 videos.

Our detailed writing principles are listed below:

**Topic**

1. Less than 15 Traditional Chinese words.
2. It must contain at least one subject, which can be a person, item, or thing

**Description**

1. Less than 150 Traditional Chinese words.
2. Use simple and direct narrative.
3. Required to be related to the content of the video.
4. Explain the topic from an objective point of view without personal opinions.

## 3.4 TOPIC AND DESCRIPTION REASONING GENERATION

In this paper, we proposed Topic and Description Reasoning Generation (TDRG) method to address the lack of explanation under the topics. This brief overview of the TDRG approach, which is shown in Fig. 1, consists of three required components and one optional component — principles, comments, large language models, and context respectively. At first, principles component is a system-level prompt to control the output format and writing style, we designed the system prompt based on our formulated principles of writing the topics and descriptions in Section 3.3. According to prior research has demonstrated the efficacy of prompt engineering as a method for assigning a role and describing desired output format to LLM (Zheng et al., 2023). Our system prompt template for principle component is shown in Fig. 2. There are role assignment and output principles to LLM in the template. Another human-level prompt component consists of comments and context, where the comments represent a clustered collection with an underlying latent topic. The context includes any additional information referenced or implied within those comments. Comments are required for the LLM, whereas the context is optional. Last but most importantly, the core component is

the large language model. Selecting an appropriate large language model significantly enhances performance in topic reasoning tasks based on comments. Generally, the more parameters an LLM has, the better its performance. We adopted zero-shot learning and fine-tuning methods to generate the topics and descriptions with LLM.

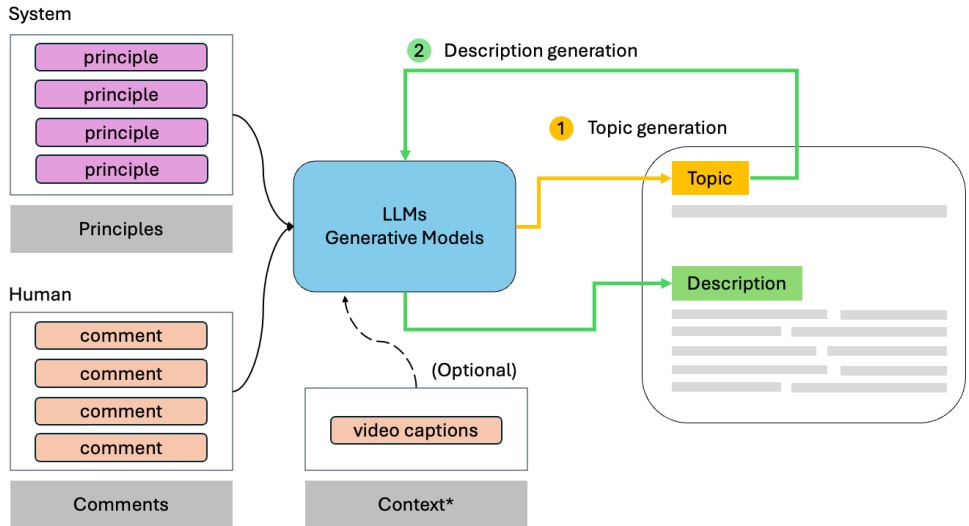

Figure 1: Overview of TDRG

**System prompt**

You are an expert in naming topic.

{role assignment and generation instruction}

Principles:

{principles list}

Figure 2: System prompt for principles

## 4  RESULT

In this section, our experiments results are evaluated through two metrics — BLEU and semantic textual similarity. **BLEU** is used to evaluate the text summarization between references sentence

and candidate sentence. **semantic textual similarity** is used to evaluate the degree of meaning similarity between two sentences, capturing the underlying semantic alignment rather than exact word matching.

We uses popular and reliable large language models, including GPT-4o and GPT-4o-mini, which were released by OpenAI in 2024. On the other hand, we also use the Llama 3.1, which is open-source AI model everyone can fine-tune, distil, and deploy anywhere. Due to our hardware restrictions, we use the Llama-3.1-8B quantized model, which may perform poorly than the original model.

## 4.1 ZERO-SHOT LEARNING LARGE LANGUAGE MODELS

In this section, we adopted the zero-shot method to generate topic and description respectively. Table. 1 and Table. 2 present the results of GPT-4o, GPT-4o-mini, Llama3.1-8B models on TDRG.

For topic generation, as shown in Table. 1, GPT-4o achieved the highest performance across all metrics, including BLEU-1 through BLEU-4 and semantic textual similarity (STS). GPT-4o-mini, a smaller variant of GPT-4o, also performed comparably well but slightly lower than GPT-4o. In contrast, Llama-3.1-8B scored lowly on all BLEU metrics and semantic similarity, indicating that it was not effective in this task under zero-shot learning.

Similarly, for description generation Table 2, GPT-4o once again outperformed the other models, achieving higher BLEU scores and a semantic textual similarity score of 0.6155. GPT-4o-mini followed closely behind, but with a small drop in BLEU and STS scores. Once again, Llama-3.1-8B performed poorly. The results indicate that the GPT-4o series, particularly GPT-4o, is significantly more effective in both topic and description generation tasks in a zero-shot setting, while the Llama-3.1-8B model struggled under these same conditions.

Table 1: Results of Topic Generation

| Metrics | GPT-4o | GPT-4o-mini | Llama-3.1-8B |
|---------|--------|-------------|--------------|
| BLEU-1  | **0.2523** | 0.2361 | 0.0460 |
| BLEU-2  | **0.1809** | 0.1613 | 0.0299 |
| BLEU-3  | **0.1241** | 0.1142 | 0.0205 |
| BLEU-4  | **0.0872** | 0.0805 | 0.0136 |
| STS     | **0.5019** | 0.4981 | 0.3964 |

Table 2: Results of Description Generation

| Metrics | GPT-4o | GPT-4o-mini | Llama-3.1-8B |
|---------|--------|-------------|--------------|
| BLEU-1  | **0.2155** | 0.1909 | 0.1340 |
| BLEU-2  | **0.1304** | 0.1121 | 0.0755 |
| BLEU-3  | **0.0790** | 0.0660 | 0.0450 |
| BLEU-4  | **0.0486** | 0.0402 | 0.0276 |
| STS     | 0.6155 | **0.6193** | 0.5555 |

## 4.2 ADD VIDEO CAPTIONS

In this section, we integrated video captions as an additional input to the large language models (LLMs) for topic and description generation. Table 3 and Table 4 show the results of the models—GPT-4o, GPT-4o-mini, and Llama-3.1-8B—on these tasks.

For topic generation with video captions, as shown in Table. 3, GPT-4o-mini performed better than GPT-4o in most BLEU metrics, particularly in BLEU-2, BLEU-3, and BLEU-4, where it exhibited stronger performance. Specifically, GPT-4o-mini achieved a BLEU-1 score of 0.1820, surpassing GPT-4o's 0.1477. GPT-4o-mini also had a higher semantic textual similarity (STS) score of 0.4439, outperforming GPT-4o, which achieved 0.4019. On the other hand, Llama-3.1-8B lagged significantly behind both GPT-4 models, with notably lower BLEU and STS scores, showing that it struggled to generate relevant topics from video captions. In the description generation task with

video captions, as shown in Table 4, GPT-4o performed better in BLEU-1 and BLEU-2 metrics, achieving scores of 0.2146 and 0.1308 respectively, while GPT-4o-mini had stronger performance in BLEU-4 and STS, scoring 0.6255 in STS compared to GPT-4o's 0.6013. Llama-3.1-8B, once again, was outperformed by both GPT-4 variants, achieving the lowest scores across all metrics, suggesting it is less effective in generating descriptions with the aid of video captions. These results demonstrate that the addition of video captions improved the models' performance in both tasks, with GPT-4o-mini showing particularly strong results in topic generation, while GPT-4o remained competitive in description generation.

Table 3: Results of Topic Generation with Video Captions

| Metrics | GPT-4o | GPT-4o-mini | Llama-3.1-8B |
|---|---|---|---|
| BLEU-1 | 0.1477 | **0.1820** | 0.0560 |
| BLEU-2 | 0.0945 | **0.1229** | 0.0357 |
| BLEU-3 | 0.0614 | **0.0843** | 0.0244 |
| BLEU-4 | 0.0408 | **0.0592** | 0.0162 |
| STS | 0.4019 | **0.4439** | 0.3633 |

Table 4: Results of Description Generation with Video Captions

| Metrics | GPT-4o | GPT-4o-mini | Llama-3.1-8B |
|---|---|---|---|
| BLEU-1 | **0.2146** | 0.1877 | 0.1266 |
| BLEU-2 | **0.1308** | 0.1113 | 0.0645 |
| BLEU-3 | **0.0805** | 0.0670 | 0.0370 |
| BLEU-4 | **0.0497** | 0.0415 | 0.0223 |
| STS | 0.6013 | **0.6255** | 0.4842 |

### 4.3 Fine-tuning Llama3.1-8B for TDRG

In this section, we adopted the fine-tuning method to generate topics and descriptions. Unlike the zero-shot approach, fine-tuning helps the model learn specific language patterns and task requirements by providing a large amount of pre-labeled training data. For the description generation sub-task, the fine-tuning method requires the model to use the output from the topic generation sub-task as context to generate descriptions more closely related to the video. This process not only handles multiple subjects that may appear in a single comment but also helps the LLM focus on a specific topic, ensuring that the description is highly relevant and accurate to the video content. Compared to the zero-shot method, fine-tuning relies on richer contextual learning and more precise semantic understanding, resulting in content that better meets specific requirements.

Our results of fine-tune model are presented in Table 5. In terms of topic generation, the BLEU scores surpass those of other models, indicating that the fine-tuned model can closely match the format of human annotations. However, the performance in description generation was less satisfactory, which might be due to the limited amount of data, resulting in the model's inability to effectively learn how to generate long texts.

Table 5: Results of Fine-tune model

| Metrics | Topic | Topic with captions | Description | Description with captions |
|---|---|---|---|---|
| BLEU-1 | 0.4023 | 0.2092 | 0.0101 | 0.0305 |
| BLEU-2 | 0.3478 | 0.1514 | 0.0060 | 0.0185 |
| BLEU-3 | 0.2998 | 0.1131 | 0.0039 | 0.0125 |
| BLEU-4 | 0.2622 | 0.0842 | 0.0025 | 0.0079 |

## 5 CONCLUSION

In this paper, we have explored topic and description reasoning generation, a novel approach to addressing the topic reasoning task from comments. We employed zero-shot learning with a large language model to generate topics and descriptions. As a result, the GPT-4o series models performed well in topic reasoning tasks due to their large parameter size. In contrast, the Llama-3.1-8B model performed poorly, often generating incorrect formats. To improve the Llama model's ability for topic reasoning, we applied the LoRA method for fine-tuning. We demonstrated the fine-tuned model can generated the correct format but still need to improve. In the future work, we will also explore the small language models abilities to topic reasoning task.

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
