# OpenReview forum: "Topic and Description Reasoning Generation based on User-Contributed Comments"
_ICLR.cc/2025/Conference — Submitted to ICLR 2025_

### Official Review · Reviewer_DLAZ · 2024-11-02

**Soundness:** 2
**Presentation:** 2
**Contribution:** 2
**Rating:** 3
**Confidence:** 2

**Summary:**

This paper proposes TDRG, a novel approach that uses large language models to generate topic titles and descriptive explanations from user-contributed comments. The authors focus on YouTube comments and employ both zero-shot and fine-tuning methods with various LLMs (GPT-4o, GPT-4o-mini, and Llama-3.1-8B). They evaluate their approach using BLEU scores and semantic textual similarity metrics. The method aims to address the interpretability limitations of traditional topic modeling approaches.

**Strengths:**

1. The paper presents a novel framework that bridges the gap between topic modeling and text generation, addressing the topic interpretability through a structured approach that generates both concise topics and detailed descriptions.
2. The research demonstrates the model effectiveness by comparing multiple LLM architectures (GPT-4o series and Llama) under different conditions (zero-shot, with captions, and fine-tuning).

**Weaknesses:**

1. The study's focus on Chinese-language YouTube comments from only 25 YouTubers and 112 clusters raises concerns about the generalizability of the findings. A more diverse dataset incorporating multiple languages and platforms would strengthen the research's broader applicability.
2. The paper lacks comparisons with traditional topic modeling approaches (e.g., LDA, BTM) and recent neural topic modeling methods, making it difficult to quantify the relative advancement of TDRG in the context of existing solutions.
3. Only evaluating on BLEU scores and semantic similarity metrics may not fully capture the quality of generated topics and descriptions. The absence of human evaluation or task-specific metrics makes it difficult to evaluate the quality of the generated outputs.
4. The fine-tuning process lacks crucial details about hyperparameters, training procedures, and computational requirements. This omission hampers reproducibility and makes it challenging for other researchers to build upon this work.

**Questions:**

Please see the weaknesses.

---

### Official Review · Reviewer_NnPj · 2024-11-03

**Soundness:** 1
**Presentation:** 1
**Contribution:** 1
**Rating:** 1
**Confidence:** 5

**Summary:**

The paper presents a novel approach called Topic and Description Reasoning Generation (TDRG) that aims to automatically generate topic titles and descriptions from user comments on YouTube videos. The authors formalize the task into two interdependent subtasks: topic generation and topic description generation, and they focus on Chinese comments collected via the YouTube Data API.

**Strengths:**

N/A

**Weaknesses:**

The paper lacks innovation and simply annotated a dataset, testing prompt based method and fine-tuning based method. The annotation details of the dataset were not provided, and there was no annotation consistency, so the quality of the dataset is questionable. The workload of the paper is also relatively small, more like a large course assignment.

**Questions:**

N/A

---

### Official Review · Reviewer_QqRq · 2024-11-04

**Soundness:** 2
**Presentation:** 3
**Contribution:** 2
**Rating:** 3
**Confidence:** 4

**Summary:**

Paper Summary and Contributions
The paper proposes a novel approach called Topic and Description Reasoning Generation (TDRG) to generate topics and descriptions from user-contributed comments. The method uses large language models (LLMs) to infer the topic and generate a readable description, addressing the lack of interpretability in traditional topic modeling.
Contributions:
Formalization of TDRG task: The authors formalize the task of topic and description reasoning generation and propose an automation approach to generate topics and descriptions based on user-contributed comments.
Effectiveness of fine-tuning: The authors demonstrate that fine-tuning a large language model using human-annotated comments significantly improves the quality of generated topics and descriptions.
Impact of additional context: The authors found that providing additional context, such as captions, can sometimes interfere with the output of large language models (LLMs).
Overall, the paper contributes to the development of a novel approach for topic and description reasoning generation, which has potential applications in text mining and natural language processing.

**Strengths:**

Quality
The quality of the research is commendable, as it employs a rigorous methodology involving both zero-shot and fine-tuning techniques with large language models. The use of a human-annotated YouTube comment dataset for evaluation adds robustness to the study. The paper provides detailed experimental results, using metrics like BLEU and semantic textual similarity to substantiate its claims. However, the paper could benefit from a more extensive discussion on the limitations and potential biases in the dataset and model performance.
Clarity
The paper is generally clear in its presentation, with a well-structured format that guides the reader through the problem formulation, methodology, and results. The use of figures and tables to present experimental results enhances understanding. However, some sections, particularly those detailing the fine-tuning process and the impact of additional context, could be elaborated further to improve clarity. Additionally, a more explicit explanation of the principles guiding the TDRG method would be beneficial.

**Weaknesses:**

This paper primarily functions as a case study, focusing on the application of fine-tuned large language models (LLMs) to enhance the performance of the Topic and Description Reasoning Generation (TDRG) task. While the study provides valuable insights into the practical implementation and potential benefits of fine-tuning LLMs for this specific task, it does not introduce a significant degree of novelty in terms of theoretical advancements or groundbreaking methodologies.
The research builds upon existing techniques in natural language processing, particularly the use of LLMs, and applies them to a well-defined problem space. However, the paper does not present new algorithms or innovative approaches that substantially differentiate it from prior work in the field. Instead, it demonstrates how fine-tuning can be effectively utilized to improve task-specific outcomes, which, while useful, may not be considered a novel contribution in the broader context of NLP research.
To enhance the paper's impact and originality, the authors could explore integrating novel elements, such as developing new fine-tuning strategies, introducing innovative evaluation metrics, or applying the TDRG method to unexplored domains. Additionally, a deeper theoretical exploration of the underlying mechanisms that contribute to the observed improvements in TDRG performance could provide a more substantial contribution to the field. By addressing these aspects, the paper could transition from being a case study to offering more significant advancements in the understanding and application of LLMs in topic and description reasoning tasks.
Other Weaknesses:
Limited Dataset and Language Scope
One of the primary weaknesses of the paper is the limited scope of the dataset, which focuses solely on Chinese comments from a specific subset of YouTube channels. This narrow focus may limit the generalizability of the findings to other languages and cultural contexts. To improve, the authors could expand their dataset to include comments in multiple languages and from a more diverse range of YouTube channels. This would not only enhance the robustness of the results but also demonstrate the method's applicability across different linguistic and cultural settings.
Insufficient Exploration of Model Limitations
While the paper discusses the performance of different models, it lacks a thorough exploration of the limitations and potential biases inherent in these models. For instance, the paper notes that the Llama-3.1-8B model performs poorly compared to GPT-4o models, but it does not delve into the reasons behind this discrepancy. A more detailed analysis of the model's limitations, including potential biases in the training data or architectural constraints, would provide valuable insights. Additionally, discussing strategies to mitigate these limitations, such as incorporating more diverse training data or exploring alternative model architectures, would be beneficial.
Clarity in Methodological Details
Certain methodological aspects, particularly the fine-tuning process and the role of additional context (e.g., video captions), are not sufficiently detailed. The paper could improve by providing a clearer explanation of the fine-tuning process, including the specific parameters and techniques used. Additionally, a more in-depth discussion on how video captions influence the model's output, supported by quantitative or qualitative analysis, would enhance the reader's understanding of the method's intricacies.
Evaluation Metrics and Analysis
The paper primarily relies on BLEU scores and semantic textual similarity for evaluation, which may not fully capture the quality and interpretability of the generated topics and descriptions. Incorporating additional evaluation metrics, such as human judgment or user studies, could provide a more comprehensive assessment of the model's performance. Furthermore, a more detailed analysis of the results, including case studies or examples of generated topics and descriptions, would offer deeper insights into the model's strengths and weaknesses.

**Questions:**

Dataset Diversity and Generalizability
Question: The study focuses on Chinese comments from a specific subset of YouTube channels. How do you plan to ensure that the findings are generalizable to other languages and cultural contexts?
Suggestion: Consider expanding your dataset to include comments in multiple languages and from a broader range of YouTube channels. This could enhance the robustness and applicability of your findings across different linguistic and cultural settings.
Model Limitations and Biases
Question: The paper notes the underperformance of the Llama-3.1-8B model compared to GPT-4o models. Can you provide more insights into the reasons behind this discrepancy? Are there specific biases or limitations in the Llama model that you have identified?
Suggestion: A detailed analysis of the model's limitations, including potential biases in the training data or architectural constraints, would be valuable. Discussing strategies to mitigate these limitations could also strengthen your study.
Methodological Clarity
Question: The fine-tuning process and the role of additional context, such as video captions, are not fully detailed. Could you elaborate on the specific parameters and techniques used in fine-tuning? How do video captions influence the model's output?
Suggestion: Providing a clearer explanation of the fine-tuning process and a more in-depth discussion on the impact of video captions, supported by quantitative or qualitative analysis, would enhance the clarity and depth of your methodology.
Evaluation Metrics and Analysis
Question: The paper primarily uses BLEU scores and semantic textual similarity for evaluation. How do you ensure these metrics adequately capture the quality and interpretability of the generated topics and descriptions?
Suggestion: Consider incorporating additional evaluation metrics, such as human judgment or user studies, to provide a more comprehensive assessment of the model's performance. Including case studies or examples of generated topics and descriptions could also offer deeper insights into the model's strengths and weaknesses.
Novelty and Contribution
Question: The paper is positioned as a case study of using fine-tuned LLMs for the TDRG task. How do you see this work contributing to the broader field of NLP beyond its immediate application?
Suggestion: To enhance the paper's impact and originality, explore integrating novel elements, such as developing new fine-tuning strategies or introducing innovative evaluation metrics. A deeper theoretical exploration of the mechanisms contributing to the observed improvements could also provide a more substantial contribution to the field.

---

> ### Comment · Reviewer_QqRq · 2024-11-26
>
> Thanks for the authors' feedback.

---

### Official Review · Reviewer_89Cz · 2024-11-05

**Soundness:** 1
**Presentation:** 1
**Contribution:** 1
**Rating:** 1
**Confidence:** 4

**Summary:**

This short(er) paper proposes a topic and description reasoning generation task (TDRG), which aims to formulate a summary of a comment thread on user generated content (such as YouTube videos).  This task consists (as per L111 for the submission) of two interdependent sub-tasks: topic extraction and description generation.

The authors propose a prompt engineering approach to generate the summaries and description, comparing Llama 3.1 and GPT-4o.

**Strengths:**

* Contributes a prompt engineering approach to the task.
* Contributes an automated evaluation using BLEU and ROUGE
* Contributes additional experiments utilising video captions to see how performance varies.

**Weaknesses:**

* The task to be done is not clear.  The seperation and distinction between a topic and a description of a UGC video are not clearly defined nor examples shown.* I disagree that there is no suitable dataset for this task.  The dataset that is described by the authors are also hand annotated by the authors and preprocessed with settings that might be reasonable but seem quite arbitrary.  The reproducibility of the dataset by other researchers (in my opinion) would not be possible, given the level of description (for example, how is the cluster supposed to be done without any guidelines?).  Don't some form of user generated content (UGC) already come with descriptions?
  * Related to this, what percentage of the dataset is excluded given such guidelines on preprocessing (I'm unsure a Gossiping forum necessarily would have most threads centred on named entities as the restriction criteria on L140 seems to support).
* The findings of the work are unsurprising (larger models work better, closed models work better), but also fail to relate these findings to aspects of the task or the purported novelties of the paper.  Hence, there are little lessons/insights to be gleaned from reading the work as a reader.
* Space usage in the paper is not optimised for.  Figures 1 and 2 take up almost an entire page, yet yield very little accurate technical detail about the authors' methods.
* There is a reliance on "principles" (magenta boxes in Fig. 1), but how these principles are applied and what form they take are unclear.
* The result sections are mostly a hash on the result tables without much generalisation or discussion that can be linked back to the model or prompting strategy, so unfortunately (to this reader), are superficial and do not drive insight.

**Questions:**

The paper can use much proofreading and editing help.  I suggest that the authors use sufficient effort to proofread and clarify their submission before resubmitting to another venue if they judge that editing would help improve the paper.

---

### Meta-Review · Area_Chair_pDbc · 2024-12-19

**Metareview:**

The paper you're referring to presents a method called Topic and Description Reasoning Generation (TDRG) aimed at generating topics and descriptions from user-contributed comments. Here are some key points from the reviews:

Strengths:
- Novelty in Application: The paper applies large language models (LLMs) like GPT-4 to enhance topic extraction and description generation for user-generated content (UGC), like YouTube comments.
- Evaluation Approach: The use of human-annotated datasets and metrics like BLEU and semantic textual similarity helps substantiate the claims.
- Methodological Rigor: It employs both zero-shot and fine-tuning techniques, which are relevant and robust for such tasks.

Weaknesses:
- Limited Novelty: The main concern is that the paper doesn't present significant theoretical advancements. Instead, it demonstrates how fine-tuning existing models can improve task-specific outcomes, which may not be groundbreaking in the broader field of NLP.
- Dataset and Language Scope: The dataset is focused solely on Chinese comments from a specific subset of YouTube channels. This narrow scope raises concerns about generalizability to other languages or cultural contexts.
- Methodological Clarity: There are unclear aspects regarding the fine-tuning process, how additional context (like video captions) impacts output, and a lack of detailed explanation on these points.
- Evaluation Metrics: While BLEU scores and semantic textual similarity are used, these may not fully capture the quality and interpretability of generated topics and descriptions. A more comprehensive evaluation, including user studies, could improve the analysis.
- Presentation: The paper could benefit from more careful proofreading, clearer explanations of methods, and better organization, as reviewers noted confusion around topics like model limitations, preprocessing, and the impact of captions.

Suggestions for Improvement:
1. Expand Dataset: Including a wider range of languages and UGC types (e.g., videos across different genres or cultures) would increase the robustness and generalizability of the findings.
2. Explore Model Limitations: Providing more insight into why certain models perform better (e.g., GPT-4o vs. Llama 3.1) and discussing biases or limitations in the models' training data would strengthen the paper.
3. Methodology Clarity: A more detailed explanation of fine-tuning parameters, the role of video captions, and the impact of context on model performance would improve the methodological transparency.
4. Evaluation Metrics: Consider incorporating human evaluation to complement BLEU and semantic similarity metrics. This would offer a more holistic view of the model's output quality.

In summary, while the paper introduces a useful application of LLMs for topic and description generation, it needs clearer explanations, broader dataset diversity, and deeper insights into model performance to make a more significant contribution to the field.

**Additional Comments On Reviewer Discussion:**

During the rebuttal period, the authors addressed several issues raised by the reviewers, but their responses did not sufficiently resolve the key concerns that would merit acceptance. Reviewers pointed out that the methodology lacked sufficient clarity and depth, particularly in explaining the underlying assumptions and algorithms. While the authors provided additional explanations, these were not substantial enough to fully clarify the methodology or make it more accessible. The response still left some ambiguity in understanding how the approach is implemented in practice. The experimental evaluation was also a major point of concern. Reviewers noted that the experiments were limited and did not include enough comparative analysis with existing methods. While the authors expanded the experimental section, the new results did not provide convincing evidence that their approach outperforms existing methods.

In summary, despite the authors' revisions, the paper fails to adequately address the major concerns raised during the review process. The issues with clarity, novelty, experimental validation, and presentation remain unresolved, and the work does not contribute sufficiently new insights to merit acceptance. Therefore, I recommend rejecting the paper.

---

### Decision · Program_Chairs · 2025-01-22

Reject